# Absolute Quantitative Volatile Measurement from Fresh Tea Leaves and the Derived Teas Revealed Contributions of Postharvest Synthesis of Endogenous Volatiles for the Aroma Quality of Made Teas

**Mingjie Chen** [1,*,†], **Li Guo** [2,3,†], **Huiwen Zhou** [1], **Yaling Guo** [3], **Yi Zhang** [1], **Zhi Lin** [2], **Meng Sun** [1], **Wei Zeng** [1] and **Hualing Wu** [4]

1   Henan Key Laboratory of Tea Plant Biology, College of Life Sciences, Xinyang Normal University, Xinyang 464000, China; zhouyihui.282@163.com (H.Z.); yitea7@163.com (Y.Z.); sunmeng2010abc@163.com (M.S.); zw@xynu.edu.cn (W.Z.)
2   Tea Research Institute, Chinese Academy of Agricultural Sciences, Hangzhou 310008, China; guoli@tricaas.com (L.G.); linz@tricaas.com (Z.L.)
3   College of Horticulture, Fujian Agriculture and Forestry University, Fuzhou 350002, China; yaling7819@126.com
4   Tea Research Institute, Guangdong Academy of Agricultural Sciences, Guangdong Key Laboratory of Tea Plant Resources Innovation & Utilization, Guangzhou 510640, China; wuhualing@163.com
*   Correspondence: mjchen@xynu.edu.cn
†   Contributed equally to this work.

**Abstract:** Characteristic aroma is a well-appreciated feature contributing to tea quality. Although extensive studies have been made to investigate aroma biosynthesis and gene expressions during tea making processes, it remains unclear whether the endogenous volatile biosynthesis during postharvest tea processing contributes to the aroma quality of made tea. To critically evaluate this question, we used the same batch of fresh tea leaves and produced three different types of tea with different degrees of fermentation (green tea, oolong tea, and black tea). Total volatiles were extracted by solvent-assisted-flavor evaporation, then quantified by gas chromatography-flame ionization detector combined with response factor correction for quantitative measurement. Compared with fresh tea leaves, the volatile profiles of the made teas were dramatically altered, with significant loss for the majority of endogenous volatiles and simultaneous gain for non-endogenous volatiles. By calculation of odor-activity values, the potential volatiles contributing to the aroma characteristics of each tea type were identified. Our data suggest that postharvest synthesis of endogenous volatiles did not contribute to the aroma quality of made tea.

**Keywords:** solvent-assisted-flavor evaporation; GC-FID; FID response factor; endogenous volatile; fresh leaves; green tea; oolong tea; black tea

## 1. Introduction

Aroma is determined by the concentration and composition of various volatiles. Tea aroma characteristics affect its quality, consumer preference, and commercial value. Based on the degrees of fermentation, tea can be classified into three large categories: non-fermented green tea, semi-fermented oolong tea, and fully fermented black tea [1,2]. Tea making starts with fresh tea leaves as the raw material, which has a grassy odor. During tea processing the volatile profiles are altered and eventually form the characteristic aroma [3–6]. Volatiles retained in the made teas are either derived from fresh tea leaves or newly synthesized during postharvest tea making. Here, for clarity, the volatiles existed in the preharvest tea leaves are named as endogenous volatiles (EVs) of fresh tea leaves. Accordingly, the non-endogenous volatiles (NEVs) refer to those that are not detected from the fresh tea leaves but present in made teas.

The quality of tea depends first and foremost, on the quality of the raw material and rational application of technological processes. EVs are affected by the tea cultivar, growing environment, and agronomic practices [7–10]. It is generally assumed that EVs synthesis during tea making processes contribute significantly to the aroma characteristics of made teas; extensive efforts have been invested to characterize EVs biosynthesis pathways and related gene expressions during postharvest tea making, in the hope to find new strategies to improve the aroma quality of made teas [11–17]. In most of these studies, volatiles were measured in a relative- or semi- quantitative manner; volatiles of fresh tea leaves versus those of made teas were not quantitatively extracted and compared. Thus, it remains unclear whether these synthesized EVs really contribute to the aroma quality of made teas. In this study, we started with the same batch of fresh tea leaves (*Camellia sinensis* cv *Tieguanyin*), and made three different types of tea (green tea, oolong tea, and black tea) according to respective standard tea processing methods (Supplementary Figure S1). Their volatiles were extracted by a solvent-assisted-flavor evaporation, then quantified by gas chromatography-flame ionization detector (GC-FID) combined with FID response factor correction. We found that compared to fresh tea leaves, the volatile profiles of made teas were dramatically altered; the majority of the EVs contents were reduced, while many NEVs accumulated simultaneously. The total volatile contents of made teas were significantly lower than those of the fresh tea leaves. By calculating the odor-activity value (OAV), the potential volatiles contributing to the aroma characteristics of each tea type were identified. Our data suggest that EVs synthesis during postharvest tea processing did not make significant contributions to the aroma quality of made tea.

## 2. Materials and Methods

### 2.1. Sample and Chemicals

One bud and three leaves were plucked from *Camellia sinensis* cv *Tieguanyin* on 29 April 2017, then processed into green tea, oolong tea, and black tea according to respective standard tea making methods (Supplementary Figure S1). The tea leaves were withered under sunlight for 30 min, then moved indoors to cool down. To make green tea, tea leaves were heat deactivated. To make oolong tea, leaves were subjected to turning-over followed by spreading. This turning-over/spreading cycle was repeated four times, the time for the first turning-over to the fourth turning-over lasted for 3-, 4-, 8-, and 10- min, respectively. The spreading lasted for 2–3 h following each turning-over treatment. After the completion of the 4th turning-over/spreading cycle, the tea leaves were heat deactivated. Heat deactivation was conducted in an electric-powered pan, the leaf temperature was monitored, and the deactivation was stopped once leaf temperature reached to 80–85 °C. The leaves were rolled for 35 min, followed by a drying treatment. To make black tea, tea leaves were withered indoors for 18 h, then rolled for one hour, followed by fermentation for 4 h before drying. The drying was performed at 100 °C for 10 min followed by 90 °C for 1 h. Before tea making, a portion of the fresh tea leaves were frozen in liquid nitrogen as the control sample.

Diethyl ether (analytical grade, 99.9%) and anhydrous sodium sulfate (analytical grade, 99%) were ordered from Sino Pharm, Shanghai, China; ethyl caprate (GC ≥ 98%) was purchased from Sigma, St. Louis, MO, USA.

### 2.2. Volatile Extraction

Tea volatiles were extracted as before [18]. The water contents from the fresh tea leaves varied between 65% to 75% depending on the tenderness of the tea leaves, germplasms and plucking seasons. In this study, 8 g of fresh tea leaves and 2 g of dry tea were used for volatile extraction. Tea leaves were frozen with liquid nitrogen and powdered, the tea powder was transferred into a glass stoppered flask. Forty milliliters of diethyl ether and 64.5 μg of ethyl caprate internal standard were added. To absorb water, 8 g (for fresh tea leaves) or 0.5 g (for dry tea) anhydrous sodium sulfate was also added. The tea powder was extracted under stirring (200 rpm) for 2 h. The organic phase was collected, then distilled

through a solvent-assisted-flavor-evaporation device (SAFE). The detailed SAFE operation followed the method described by Chen et al. (2020) [18]. The volume for the distillation flask and the receiver flask was 500 mL. The volatile distillate was collected, aliquot into 8-mL glass tubes, concentrated in a rotary evaporator to about 1.5 mL, then transferred into labeled 2-mL GC vials, and further concentrated in a rotary evaporator. The volume was closely monitored; the concentration step was stopped immediately once the volume was slightly below the 500 µL marker line. A small volume of diethyl ether was added to make the total volume reach 500 µL. Fresh leaves and oolong tea had four biological replicates, whereas green tea and black tea had three biological replicates.

### 2.3. Volatile Identification and Quantification

Volatiles were analyzed as described before [18,19]. Briefly, 1 µL of sample was injected into a capillary column (RXi-5SiIMS column, 30 m × 0.25 mm × 0.25 µm) in a splitless mode for GC-MS and GC-FID analysis. The oven temperature program was initiated at 50 °C for 2 min, raised 5 °C min$^{-1}$ to 180 °C, and held 2 min; then raised 10 °C min$^{-1}$ to 230 °C, and held 5 min before returning to 50 °C for the next sample. Injector, ion source, interface, and FID temperatures were 250, 230, 270, and 230 °C, respectively. The flow rates of hydrogen, nitrogen, and zero air were 40, 30, and 400 mL min$^{-1}$, respectively, and helium was used as the carrier gas with a flow rate of 1.0 and 1.7 mL min$^{-1}$ for GC-MS and GC-FID, respectively. Zero air is a highly purified air in which almost all the hydrocarbon is removed. The electron ion source was set at 70 eV, and the chromatograms were recorded by monitoring the total ion current in the mass range of 45–600 m/z. Since each compound has its own ionization and fragmentation in the MS detector [20], thus, various volatiles cannot be reliably quantified by comparing with a single internal standard. In this study, MS data was used for volatile identification by comparing MS spectral with the NIST14 database, and further confirmed by their reported retention index; volatiles were quantified from FID data by normalizing to the internal standard peak area. To calibrate the FID response differences for individual volatiles, FID response factors relative to the internal standard were calculated and used for further normalization (Supplementary Table S1). The fresh leaf water contents were determined by the National Standard of P. R. China (GB/T 5009-2016), then used to convert fresh leaf weight into dry mass. The volatile contents were normalized to dry mass. To convert the retention time into the retention index, n-alkanes (C9–C25) were analyzed under the same condition, and the data was used to convert the retention time into the retention index (RI) by the formula: RI = 100 × [(tR − tRz)/(tR(z + 1) − tRz) + z]. Where z is the carbon number of the alkane standard, tRz and tR(z + 1) represent the retention times of z and z + 1 alkane standards, and tR is the retention time of tea volatile that falls between tRz and tR(z + 1).

### 2.4. Calculation of Volatile Concentration from Tea Infusion and Odor-Activity Value

Organoleptic assessment was made according to the National Standard of P. R. China (GB/T 23776-2018)—Methodology for Sensory Evaluation of Tea. In a standard organoleptic assessment by a trained tea taster, the oolong tea infusion was prepared by extracting 5 g of oolong dry tea in 110 mL of boiling water; in contrast, the green tea or black tea infusion was prepared by extracting 3 g of dry tea in 150 mL of boiling water. To evaluate the volatile contribution to the aroma characteristics of tea infusion, ideally their concentrations in the tea infusion are directly measured. However, to fully recover volatiles from the simulated standard tea infusion preparation is technically challenging, since a considerable amount of volatiles easily evaporate into the air space and get lost; to add this complexity, it has been reported that during tea infusion preparation, under the action of hot water, some new volatiles are formed from its glycosidically bound precursors [21]. To avoid these issues, we assumed that all the volatiles from dry tea were extracted into the tea infusion during the tea infusion preparation. Based on the absolute volatile contents from dry tea and the tea mass to water ratio, the volatile concentrations in the tea infusion could be

calculated. The odor detection threshold in water was obtained from literature, then used to calculate odor activity value.

### 2.5. Statistical Analysis

A Student's t-test with two-tailed equal variance was performed to determine the significance ($p < 0.05$).

### 3. Results and Discussion

The volatiles from fresh tea leaves, green tea, oolong tea, and black tea were extracted in diethyl ether, then separated from non-volatile substances and analyzed by GC-MS and GC-FID (Supplementary Figures S2 and S3). In total; 31, 32, 27, and 31 volatiles were identified from respective samples (Table 1). Among the 31 EVs of fresh tea leaves, there were 19, 15, and 14 volatiles detected from green tea, oolong tea, and black tea, respectively. Among the 32 green tea volatiles, there were 17 and 15 volatiles commonly detected from oolong tea and black tea, respectively. There were 14 volatiles commonly detected between oolong tea and black tea (Table 1, Supplementary Figure S4).

Among the 31 EVs detected from fresh tea leaves, nine volatiles were below the detection threshold from these three types of made tea: phenylethyl alcohol, germacrene D, nerolidol isobutyrate, γ-cadinene, trans-calamenene, *cis*-cadina-1(2),4-diene, and three unidentified volatiles. Meanwhile, nine volatiles from fresh tea leaves were commonly detected from these three types of made teas. In four of them, β-*cis*-ocimene, *cis*-linalool oxide (furanoid), *trans*-linalool oxide (furanoid), and linalool, their contents from made teas only accounted for 0.8–14% of that from fresh tea leaves. The other five volatiles, including *E*-nerolidol, (*Z*)- 3-hexenyl benzoate, hexyl benzoate, neophytadiene, and caffeine, were not significantly different from that of fresh tea leaves except that the *E*-nerolidol contents from green tea and oolong tea as well as the neophytadiene and caffeine contents from black tea were significantly higher than that of the fresh tea leaves. The remaining 13 volatiles were either detected from only one or two types of made teas; their contents were significantly lower than that of fresh tea leaves except *cis*-jasmone and β-caryophyllen, which were similar between fresh tea leaves and green tea or oolong tea (Table 1). The total volatile contents for fresh tea leaves, green tea, oolong tea, and black tea were 76.13 ± 11.66, 7.73 ± 0.71, 5.77 ± 0.74, 7.80 ± 1.83 μg. g$^{-1}$ dry weight (DW), respectively. These data demonstrated that the tea making processes were characterized by the significant loss for most of the EVs. Thus, their synthesis during postharvest tea making processes would not contribute significantly to the aroma quality of made tea products. It remains unclear why and how these EVs were lost during tea making processes. A detailed quantitative volatile measurement at each processing step is required to resolve this issue. It is obvious that the abundant volatiles from fresh tea leaves were lost more compared to less abundant volatiles (Table 1). We speculated that there are two factors that could affect volatile retention. One was that the leaf cell structure was disrupted by the tea making process, which may reduce the leaf's volatile holding capacity, and thus result in the evaporation of the abundant volatiles. On the other hand, drying should result in significant volatile loss since it was performed at 90–100 °C, which was above the boiling points for many tea volatiles. Under this pathway, the amounts of volatile loss would depend on the drying temperature, time, and the volatility of individual tea aroma.

**Table 1.** Volatile contents from fresh tea leaves, green tea, oolong tea, and black tea.

| RT (min) | RI$_{lit}$ | RI$_{exp}$ | Identification | Boiling Point (°C) | Aroma Descriptions | Contents (µg. g$^{-1}$ DW) | | | |
|---|---|---|---|---|---|---|---|---|---|
| | | | | | | Fresh Leaves (*n* = 4) | Green Tea (*n* = 3) | Oolong Tea (*n* = 4) | Black Tea (*n* = 3) |
| 8.966 | | 960 | unknown 1 | nf | nf | 2.45 ± 1.19 | nd | nd | nd |
| 9.344 | | 983 | unknown 2 | nf | nf | 6.47 ± 2.05 | nd | nd | nd |
| 10.31 | | 1022 | unknown 3 | nf | nf | 1.26 ± 0.27 | nd | nd | nd |
| 10.631 | 1028.9 | 1031 | β-cis-ocimene | 175.2 | Fruity, floral | 4.71 ± 1.48 [a] | 0.10 ± 0.03 [b] | 0.11 ± 0.03 [b] | 0.05 ± 0.01 [c] |
| 11.388 | 1075.1 | 1079.4 | cis-linalool oxide (furanoid) | 188 | Fresh, floral | 3.55 ± 0.51 [a] | 0.12 ± 0.03 [b] | 0.12 ± 0.03 [b] | 0.41 ± 0.10 [c] |
| 11.876 | 1083.3 | 1080.1 | trans-linalool oxide (furanoid) | 222.6 | flower | 5.94 ± 0.98 [a] | 0.13 ± 0.04 [b] | 0.13 ± 0.03 [b] | 0.86 ± 0.21 [c] |
| 12.272 | 1099 | 1095.7 | Linalool | 199 | flower, lavender | 23.14 ± 2.09 [a] | 0.23 ± 0.06 [b] | 0.19 ± 0.04 [b] | 1.06 ± 0.27 [c] |
| 12.427 | 1106.8 | 1104 | Hotrienol | 80–84 | Fruity | nd | nd | 0.42 ± 0.07 [a] | nd |
| 12.691 | 1113 | 1109 | (E)-4,8-dimethylnona-1,3,7-triene | 81 | nf | 3.47 ± 0.81 [a] | 0.14 ± 0.03 [b] | nd | nd |
| 13.383 | 1128 | 1124.3 | Benzyl nitrile | 190.7 | pungent smell | 0.32 ± 0.03 [a] | 0.17 ± 0.04 [b] | 0.20 ± 0.08 [b] | nd |
| 14.475 | 1114.9 | 1113.1 | Phenylethyl alcohol | 219 | Fresh, rose aroma | 0.14 ± 0.02 [a] | nd | nd | nd |
| 14.604 | 1164.7 | 1162 | (Z)-3-Hexenyl butanoate | 98 | Aroma like apple and grass | 0.45 ± 0.09 [a] | 0.27 ± 0.06 [b] | nd | 0.22 ± 0.07 [b] |
| 14.677 | 1171 | 1174 | trans-pyranoid linalool oxide | 201–202 | Earthy | nd | nd | nd | 0.41 ± 0.07 [a] |
| 14.865 | 1180 | 1178 | Dodecane | 216 | nf | 0.88 ± 0.19 [a] | nd | 0.10 ± 0.02 [b] | 0.14 ± 0.03 [b] |
| 15.107 | 1187 | 1184.1 | Methyl salicylate | 223.3 | Minty flavor | 11.04 ± 1.51 [a] | 0.08 ± 0.02 [b] | nd | 0.45 ± 0.12 [c] |
| 15.386 | 1189 | 1188 | 2,6-dimethyl-3,7-octadiene-2,6-diol | 284 | pungent and bad smell | nd | nd | 0.11 ± 0.03 [a] | nd |
| 16.231 | 1191.5 | 1190 | Hexyl butanoate | 205 | Green, fruity | 0.56 ± 0.10 [a] | nd | 0.08 ± 0.02 [b] | 0.08 ± 0.03 [b] |
| 16.252 | 1200.9 | 1203 | 4-methylpentyl 2-methylbutanoate | nf | nf | nd | nd | nd | 0.12 ± 0.03 [a] |
| 16.418 | 1236.3 | 12,331 | Hexyl 2-methyl butanoate | 217–219 | Green, fruity | nd | nd | nd | 0.11 ± 0.03 [a] |

**Table 1.** *Cont.*

| RT (min) | RI$_{lit}$ | RI$_{exp}$ | Identification | Boiling Point (°C) | Aroma Descriptions | Contents (µg. g$^{-1}$ DW) | | | |
|---|---|---|---|---|---|---|---|---|---|
| | | | | | | Fresh Leaves (*n* = 4) | Green Tea (*n* = 3) | Oolong Tea (*n* = 4) | Black Tea (*n* = 3) |
| 16.899 | 1253 | 1250 | (E)- geraniol | 230 | rose, floral | 2.30 ± 0.77 [a] | 0.14 ± 0.06 [b] | nd | nd |
| 17.22 | 1249 | 1254 | 1,3-bis(1,1-dimethylethyl)benzene | 106–107 | nf | nd | nd | 0.09 ± 0.01 | nd |
| 17.595 | 1260 | 1261 | 2,5-dihydro-2,5-dimethoxyfuran | 160–162 | nf | nd | nd | 0.12 ± 0.02 | nd |
| 18.033 | 1298.4 | 1301.6 | Indole | 253–254 | mothball, burnt | 5.19 ± 0.81 [a] | 0.73 ± 0.15 [b] | 0.78 ± 0.18 [b] | nd |
| 20.516 | 1333 | 1330 | cis-3-hexenyl hexanoate | 115 | Tender, fresh and clean aroma | 1.05 ± 0.14 [a] | 0.26 ± 0.06 [b] | nd | nd |
| 20.679 | 1368 | 1370 | trans-2-hexenyl hexanoate | 79 | fruity | nd | 0.23 ± 0.05 [a] | 0.20 ± 0.02 [a] | 0.51 ± 0.11 [b] |
| 20.753 | 1381 | 1378 | Hexyl hexanoate | 246 | fruity | nd | 0.06 ± 0.02 [a] | 0.12 ± 0.03 [b] | 0.19 ± 0.04 [c] |
| 20.837 | | 1382 | unknown 4 | nf | nf | nd | nd | nd | 0.10 ± 0.11 |
| 21.124 | 1394.6 | 1396.8 | cis-Jasmone | 248 | sweet, flower | 0.15 ± 0.01 | 0.18 ± 0.03 | 0.18 ± 0.04 | nd |
| 21.743 | 1413.5 | 1410.7 | β-caryophyllen | 262–264 | nf | 0.45 ± 0.07 | 0.43 ± 0.66 | nd | nd |
| 21.751 | 1420.1 | 1418 | (E)-caryophyllene | 266–268 | nf | nd | nd | nd | 0.05 ± 0.01 [a] |
| 22.36 | 1453.1 | 1451 | α-humulene | nf | nf | nd | nd | nd | 0.06 ± 0.01 [a] |
| 22.361 | 1455.9 | 1451.6 | (E)-β-farnesene | 271 | Floral | nd | nd | 0.01 ± 0.03 [a] | nd |
| 22.468 | 1468.2 | 1463.3 | (E)-2-dodecenal | 93 | nf | nd | nd | nd | 0.05 ± 0.00 [a] |
| 22.696 | 1472.8 | 1472 | 1-dodecanol | 255–259 | grease | nd | nd | nd | 0.07 ± 0.03 [a] |
| 23.367 | 1480.6 | 1478.6 | Germacrene D | 279.7 | floral | 0.11 ± 0.02 [a] | nd | nd | nd |
| 23.382 | 1485.9 | 1484 | trans-β-ionone | 239 | seaweed, violet, flower, raspberry | nd | 0.17 ± 0.17 [a] | nd | 0.06 ± 0.01 [a] |
| 23.455 | 1504.1 | 1499.4 | α-farnesene | 260 | Floral, fresh | nd | 0.31 ± 0.23 [a] | 0.39 ± 0.03 [a] | nd |
| 23.471 | | 1502 | Nerolidol isobutyrate | 361.9 | Sweet, rose-like | 0.23 ± 0.08 [a] | nd | nd | nd |
| 23.482 | 1510 | 1507 | Tridecanal | 132–136 | nf | nd | nd | nd | 0.16 ± 0.02 [a] |
| 23.73 | 1513.1 | 1512 | γ-cadinene | 272 | Floral, fresh, sweet | 0.05 ± 0.06 [a] | nd | nd | nd |
| 23.9 | 1517.8 | 1518 | cis-jasmin lactone | 130 | sweet, flower | nd | 0.07 ± 0.01 [a] | nd | nd |

**Table 1.** *Cont.*

| RT (min) | RI$_{lit}$ | RI$_{exp}$ | Identification | Boiling Point (°C) | Aroma Descriptions | Contents (µg. g$^{-1}$ DW) | | | |
|---|---|---|---|---|---|---|---|---|---|
| | | | | | | Fresh Leaves (*n* = 4) | Green Tea (*n* = 3) | Oolong Tea (*n* = 4) | Black Tea (*n* = 3) |
| 24.205 | 1523.2 | 1520 | δ-cadinene | 279 | nf | nd | nd | 0.09 ± 0.01 [a] | nd |
| 24.311 | 1528 | 1524 | trans-calamenene | 285–286 | nf | 0.21 ± 0.04 [a] | nd | nd | nd |
| 24.323 [b] | 1529 | 1526 | Naphthalene | 217.9 | Tar, camphoric and greasy odor | nd | 0.12 ± 0.02 [a] | nd | 0.09 ± 0.01 [b] |
| 24.431 | | 1527.2 | cis-hexahydro-8a-methyl-1,8(2H,5H)-naphthalenedione | nf | nf | nd | 0.08 ± 0.02 | 0.07 ± 0.00 | nd |
| 24.637 | 1531 | 1532 | Cis-cadina-1(2),4-diene | 137.9 | nf | 0.11 ± 0.04 [a] | nd | nd | nd |
| 24.644 | 1532 | 1533.4 | (E)-γ-bisabolene | 262 | nf | nd | 0.03 ± 0.00 [a] | nd | nd |
| 24.998 | 1535 | 1536 | Dihydroactinidiolide | 296.1 | Floral, rose-like | nd | nd | nd | 0.11 ± 0.06 [a] |
| 25.149 | 1533.3 | 1537.3 | α-cadinene | 271 | green | 0.08 ± 0.02 [a] | 0.04 ± 0.01 [b] | nd | nd |
| 25.308 [c] | 1541 | 1543 | E-nerolidol | 276 | Slight neroli-like, rose-like and sweet flavor | 0.47 ± 0.14 [a] | 1.17 ± 0.10 [b] | 1.03 ± 0.18 [b] | 0.51 ± 0.08 [c] |
| 25.324 | 1550.9 | 1547 | Germacrene B | 287.2 | nf | nd | 0.11 ± 0.03 [a] | nd | nd |
| 25.626 [c] | 1572.9 | 1568 | (3E,7E)-4,8,12-trimethyltrideca-1,3,7,11-tetraene | 293.2 | nf | 0.52 ± 0.09 [a] | nd | 0.13 ± 0.01 [b] | 0.20 ± 0.04 [c] |
| 25.773 | 1562 | 1570 | cis-3-hexanyl n-octanoate | 292.5 | nf | nd | nd | 0.14 ± 0.02 [a] | nd |
| 25.928 | 1569.5 | 1573 | (Z)-3-hexenyl benzoate | 105 | Green, herb-like | 0.06 ± 0.08 | 0.08 ± 0.01 | 0.08 ± 0.00 | 0.07 ± 0.01 |
| 26.297 | 1581.8 | 1580 | Hexyl benzoate | 272 | Wood, green | 0.07 ± 0.05 [a] | 0.11 ± 0.01 [a] | 0.13 ± 0.01 [b] | 0.14 ± 0.04 [b] |
| 26.667 | 1588 | 1590.1 | E-2-hexenyl benzoate | 165–166 | nf | nd | 0.04 ± 0.00 [a] | nd | nd |
| 27.048 | 1655 | 1650 | Methyl jasmonate | 302.9 | Sweent, jasmine-like | nd | 1.04 ± 0.74 [a] | nd | nd |
| 27.676 | 1670 | 1672 | cis-3-hexenyl salicylate | 145 | Flower, green | nd | 0.08 ± 0.01 [a] | nd | nd |
| 27.677 [a] | 1676.3 | 1673.2 | 1-tetradecanol | 289 | Wax-like | nd | nd | nd | 0.06 ± 0.01 [a] |
| 28.337 | 1673 | 1675 | α-cadinol | 137–139 | Sweet, baking aroma | nd | nd | nd | 0.05 ± 0.00 [a] |
| 29.903 [b] | 1827 | 1823.1 | Neophytadiene | 128 | nf | 0.09 ± 0.04 [a] | 0.07 ± 0.02 [a] | 0.06 ± 0.01 [a] | 0.41 ± 0.19 [b] |
| 31.359 | 1830 | 1834 | Hexadecanal | 151 | nf | nd | nd | nd | 0.05 ± 0.02 [a] |
| 32.162 [b] | 1842 | 1840 | Caffeine | 178 | nf | 0.64 ± 0.07 [a] | 0.85 ± 0.20 [a] | 0.62 ± 0.39 [a] | 0.94 ± 0.20 [b] |
| 32.985 | 1927 | 1931 | n-hexadecanoic acid methyl ester | 185 | nf | nd | 0.09 ± 0.02 [a] | 0.05 ± 0.04 [a] | nd |
| Total | | | | | | 76.13 ± 11.66 [a] | 7.73 ± 0.71 [b] | 5.77 ± 0.74 [c] | 7.80 ± 1.83 [b] |

nf: not found; nd: not detected; RT: retention time in minutes; RI$_{lit}$: retention index from literature; RI$_{exp}$: retention index from this study. The different superscript letters from the same row indicate statistically significant ($p < 0.05$). Data are expressed as mean ± standard deviation.

A major difference between green tea and oolong tea making technology was the "turning over" step of oolong tea (Supplementary Figure S1). Some volatile contents from oolong tea, including hotrienol, dodecane, 2,6-dimethyl-3,7-octadiene-2,6-diol, hexyl butanoate, 1,3-bis(1,1-dimethylethyl) benzene, 2,5-dihydro-2,5-dimethoxyfuran, hexyl hexanoate, (*E*)-β-farnesene, δ-cadinene, (3E,7E)-4,8,12-trimethyltrideca-1,3,7,11-tetraene, and *cis*-3-hexanyl n-octanoate, were significantly higher than that of green tea (Table 1). This suggested that the "turning over" treatments could activate these volatile syntheses. Meanwhile, other volatile contents from oolong tea, including (*E*)-4,8-dimethylnona-1,3,7-triene, (*Z*)-3-hexenyl butanoate, methyl salicylate, (*E*)-geraniol, *cis*-3-hexenyl hexanoate, β-caryophyllen, trans-β-ionone, *cis*-jasmin lactone, naphthalene, (*E*)-γ-bisabolene, α-cadinene, germacrene B, E-2-hexenyl benzoate, methyl jasmonate, and *cis*-3-hexenyl salicylate, were lower than that of green tea, suggesting that the "turning over" step could also facilitate these volatile's evaporation.

There were 13, 12, and 17 volatiles detected from green tea, oolong tea, and black tea respectively, but below the detection limit from fresh tea leaves (Table 1, Supplementary Figure S2). These data demonstrated that postharvest tea processing facilitate NEVs formation and accumulation. Interestingly, oolong tea was generally regarded to be more aromatic than green tea or black tea [22]. However, its total aroma contents were significantly lower than that of green tea or black tea (Figure 1). Thus, the total volatile contents did not positively correlate with their aroma quality. Other factors, such as the volatile compositions, could play a larger role to influence tea's aroma characteristics.

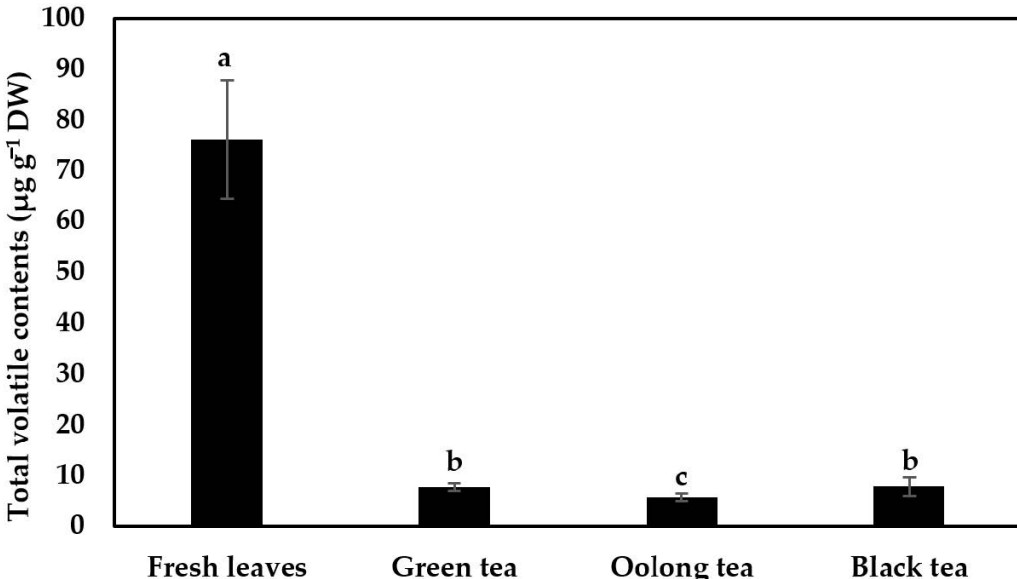

**Figure 1.** Total volatile contents from fresh leaves, green tea, oolong tea, and black tea. Data are expressed as average ± standard deviation. Different letters indicates statistically significant ($p < 0.05$).

To identify which volatiles could confer an aroma note in a standard sensory evaluation, the odor-activity value (OAV) was calculated. The odor activity values (OAVs) are the ratios of concentration found in the food matrix to its odor detection threshold [23]. The odor detection threshold is the lowest concentration of a certain odor compound that is perceivable by the human sense of smell. The OAV is commonly used to estimate odor potency, their values from fresh tea leaves and made teas are listed in the order of retention time from low to high in Table 2. The volatiles with OAV > 1 were generally regarded as contributors to the aroma characteristics. Thus, linalool, (*E*)-geraniol, indole, *cis*-jasmone, *trans*-β-Ionone, and α-farnesene could be the potential contributors to the aroma quality of *Tieguanyin* green tea; linalool, indole, *cis*-jasmone, and α-farnesene could be the potential contributors to the aroma quality of *Tieguanyin* oolong tea; linalool, *trans*-linalool oxide

(pyranoid), 1-dodecanol, and *trans*-β-ionone could be the potential contributors to the aroma quality of *Tieguanyin* black tea. Interestingly, among these potential aroma contributors, linalool, (*E*)-geraniol, and indole are EVs. Although their contents in the made teas were significantly lower than that of fresh tea leaves, they still retained a sufficient amount contributing to the aroma characteristics in the made tea. On the other hand, *tran*-β-ionone, α-farnesene, *trans*-linalool oxide (pyranoid), and 1-dodecanol, which were NEVs produced in sufficient amount during tea processing, become aroma contributors.

Although a 100% volatile recovery from dry tea to tea infusion is not plausible, considering that most of these OAVs were larger than two, thus, even with a 50% recovery rate, the OAVwould still be larger than one, and contributed to the characteristic aroma of tea infusion. Although linalool, (*E*)-geraniol, and indole were present in fresh leaves at much higher contents compared to made teas (Table 1), the fresh tea leaves did not show the pleasant aroma as the made tea did. One possibility could be that fresh tea leaves also contained other unpleasant volatiles which created masking effects. Removing or reducing these endogenous unpleasant volatiles could be an important aspect for tea processing; another possibility could be that some volatiles showed totally different characteristics in a concentration-dependent manner. One well-characterized example was indole; at very low concentrations indole had a flowery smell [39], however, in high concentrations it smelled animal-like [40]. Under our experimental conditions, only about 14% of the indole from fresh tea leaves was retained in green tea or oolong tea (Table 1). However, their OAV values in green tea and oolong still were in the ranges of 14.7–35.2 (Table 2), thus, it could confer a flowery smell to green tea or oolong tea. Four contributing volatiles, including *trans*-pyranoid linalool oxide, 1-dodecanol, *trans*-β-ionone, and α-farnesene were only detected from certain types of made tea and with large difference in odor-activity value, suggesting that tea processing technologies determined specific NEVs formation and retention, and thus shaped the characteristic aroma for each tea type.

Previously Zhang et al. (2013) applied simultaneous distillation extraction method and isolated volatiles from green teas, oolong teas, and black teas with various origins, then semi-quantified by two-dimensional GC-TOFMS, through multivariate data analysis the compounds with a significant difference were defined [41]. Several volatiles were commonly detected from these two studies; they can be divided into two groups. The first group showed similar changing trends among green tea, oolong tea and black tea between these two studies, including cis-linalool oxide (furanoid), trans-linalool oxide (furanoid), benzyl nitrile, methyl salicylate, 2,6-dimethyl-3,7-octadien-2,6-diol, indole, cis-jasmone, α-farnesene, and nerolidol; in contrast, the second group showed different changing trends among green tea, oolong tea, and black tea between these two studies, including (E)-Hotrienol, (E)-4,8-dimethyl-nona-1,3,7-triene, phenylethyl alcohol, cis-3-hexenyl hexanoate, jasmin lactone, and methyl jasmonate. Since in the previous study green tea, oolong tea and black tea came from different sources, and different cultivars may be used to produce them, this could account for the difference between these two studies. To support this notion, even within same tea type large variations have been reported in previous study [41].

**Table 2.** Aroma contents from tea infusion and odor-activity value (OAV) of green tea, oolong tea, and black tea.

| Compound | Threshold (µg. L$^{-1}$) | Ref. | Concentration in Tea Infusion (µg. L$^{-1}$) | | | OAV | | |
|---|---|---|---|---|---|---|---|---|
| | | | Green Tea | Oolong Tea | Black Tea | Green Tea | Oolong Tea | Black Tea |
| β-cis-ocimene | 34 | [24] | 2.0 | 5.1 | 1.0 | <1 | 0<1 | <1 |
| cis-linalool oxide (furanoid) | 320 | [25] | 2.4 | 5.3 | 8.2 | <1 | <1 | <1 |
| trans-linalool oxide (furanoid) | 320 | [25] | 2.7 | 6.1 | 17.2 | <1 | <1 | <1 |
| Linalool | 1 | [24] | 4.6 | 8.9 | 21.3 | 4.6 | 8.9 | 21.3 |
| Hotrienol | 110 | [26] | nd | 18.9 | nd | <1 | <1 | <1 |
| (E)-4,8-dimethylnona-1,3,7-triene | nf | | 2.8 | nd | nd | nf | nf | nf |
| Benzyl nitrile | 1000 | [25] | 3.3 | 9.3 | nd | <1 | <1 | <1 |
| (Z)-3-hexenyl butanoate | 500 | [24] | 5.4 | nd | 4.4 | <1 | <1 | <1 |
| trans- linalool oxide (pyranoid) | 0.025 | [27] | nd | nd | 8.2 | <1 | <1 | 329.0 |
| Dodecane | nf | | nd | 4.4 | 2.8 | nf | nf | nf |
| Methyl salicylate | 40 | [28] | 1.5 | nd | 9.0 | <1 | <1 | <1 |
| 2,6-dimethyl-3,7-octadiene-2,6-diol | 89 | | nd | 5.2 | nd | <1 | <1 | <1 |
| Hexyl butanoate | 607 | [29] | nd | 3.8 | 1.6 | <1 | <1 | <1 |
| 4-methylpentyl 2-methylbutanoate | nf | | nd | nd | 2.3 | nf | nf | nd |
| Hexyl 2-methyl butanoate | nf | | nd | nd | 2.2 | nf | nf | nf |
| (E)- geraniol | 1.1 | [30] | 2.8 | nd | nd | 2.6 | <1 | <1 |
| 1,3-bis(1,1-dimethylethyl)benzene | 81 | | nd | 3.9 | nd | <1 | <1 | <1 |
| 2,5-dihydro-2,5-dimethoxyfuran | 13 | [24] | nd | 5.5 | nd | <1 | <1 | <1 |
| Indole | 1 | [31] | 14.7 | 35.3 | nd | 14.7 | 35.2 | <1 |
| cis-3-hexenyl hexanoate | 16 | [25] | 5.1 | nd | nd | <1 | <1 | <1 |
| trans-2-hexenyl hexanoate | 781 | [32] | 4.7 | 9.1 | 10.1 | <1 | <1 | <1 |
| Hexyl hexanoate | 820 | [32] | 1.1 | 5.6 | 3.9 | <1 | <1 | <1 |

**Table 2.** *Cont.*

| Compound | Threshold (µg. L$^{-1}$) | Ref. | Concentration in Tea Infusion (µg. L$^{-1}$) | | | OAV | | |
|---|---|---|---|---|---|---|---|---|
| | | | Green Tea | Oolong Tea | Black Tea | Green Tea | Oolong Tea | Black Tea |
| unknown 4 | nf | | nd | nd | 2.0 | nf | nf | nf |
| cis-Jasmone | 1.9 | [33] | 3.7 | 8.4 | nd | 1.9 | 4.4 | <1 |
| β-caryophyllen | 150 | [24] | 8.5 | nd | nd | <1 | <1 | <1 |
| (E)-caryophyllene | 64 | [34] | nd | nd | 0.9 | <1 | <1 | <1 |
| α-humulene | 390 | [24] | nd | nd | 1.2 | <1 | <1 | <1 |
| (E)-β-farnesene | 87 | | nd | 0.6 | nd | <1 | <1 | <1 |
| (E)-2-dodecanal | 20 | [25] | nd | nd | 1.1 | <1 | <1 | <1 |
| 1-dodecanol | 0.5 | [35] | nd | nd | 1.4 | <1 | <1 | 2.7 |
| trans-β-ionone | 0.2 | [25] | 3.4 | nd | 1.1 | 16.9 | <1 | 5.7 |
| α-farnesene | 0.1 | [36] | 6.1 | 17.8 | nd | 61.0 | 178 | <1 |
| γ-cadinene | nf | | 1.4 | nd | nd | nf | nf | nf |
| cis-jasmin lactone | 2000 | [37] | nd | nd | 3.2 | <1 | <1 | <1 |
| δ-cadinene | 120 | [37] | nd | 4.1 | nd | <1 | <1 | <1 |
| trans-calamenene | nf | | 2.4 | nd | 1.8 | nf | nf | nf |
| Naphthalene | 300 | | 1.6 | 3.1 | nd | <1 | <1 | <1 |
| 8a-methylhexahydro-1,8(2H,5H)-naphthalenedione | nf | | nd | nd | 2.3 | nf | nf | nf |
| (R)-4,4,7a-trimethyl-5,6,7,7a-tetrahydrobenzofuran-2(4H)-one | nf | | 0.8 | nd | nd | nf | nf | nf |
| α-cadinene | nf | | 0.7 | nd | nd | nf | nf | nf |
| E-nerolidol | 15 | [38] | 2.27 | nd | nd | <1 | <1 | <1 |
| Germacrene B | nf | | 23.4 | 47.0 | 10.2 | nf | nf | nf |
| (3E,7E)-4,8,12-trimethyltrideca-1,3,7,11-tetraene | nf | | nd | 6.0 | 4.0 | nf | nf | nf |
| cis-3-hexanyl n-octanoate | nf | | nd | 6.2 | nd | nf | nf | nf |
| (Z)-3-hexenyl benzoate | 4.5 | | 1.6 | 3.6 | 1.4 | <1 | <1 | <1 |
| Hexyl benzoate | nf | | 2.2 | 6.0 | 2.9 | nf | nf | nf |

**Table 2.** *Cont.*

| Compound | Threshold (µg. $L^{-1}$) | Ref. | Concentration in Tea Infusion (µg. $L^{-1}$) | | | OAV | | |
|---|---|---|---|---|---|---|---|---|
| | | | Green Tea | Oolong Tea | Black Tea | Green Tea | Oolong Tea | Black Tea |
| E-2-hexenyl benzoate | nf | | nd | nd | 1.2 | nf | nf | nf |
| Methyl jasmonate | 70 | [38] | nd | nd | 0.9 | <1 | <1 | <1 |
| cis-3-hexenyl salicylate | nf | | 0.9 | nd | nd | nf | nf | nf |
| 1-tetradecanol | 5000 | [37] | 20.8 | nd | nd | <1 | <1 | <1 |
| α-cadinol | nf | | 1.7 | nd | nd | nf | nf | nf |
| Neophytadiene | nf | | 1.4 | 2.6 | 8.3 | nf | nf | nf |
| Hexadecanal | 75 | [25] | nd | nd | 1.1 | <1 | <1 | <1 |
| Caffeine | 29,000 | [39] | 17.0 | 28.3 | 18.8 | <1 | <1 | <1 |
| n-hexadecanoic acid methyl ester | 1000 | [25] | 1.8 | 2.3 | nd | <1 | <1 | <1 |

nd: not detected; nf: not found.

## 4. Conclusions

In this study, we successfully produced green tea, oolong tea, and black tea from the same batch of fresh leaves by following respective standard tea processing methods; tea volatiles were then extracted and quantified by GC-FID, further calibrated by FID response factor, and compared with that of fresh leaves. We found that total volatile contents from made teas were significantly reduced compared to the contents of fresh tea leaves; most of the EVs from made teas were lower in amounts than those of fresh tea leaves. Meanwhile, specific NEVs were produced during tea making processes. Calculation of odor-activity values identified that some EVs and NEVs both contributed to the characteristic aroma of made teas. Our data suggest that the postharvest endogenous volatile synthesis during tea making processes were not important to shape the characteristic aroma of made tea.

**Supplementary Materials:** The following are available online at https://www.mdpi.com/2076-3417/11/2/613/s1, Figure S1: Diagram presentation of standard tea making processes for green tea, oolong tea, and black tea, Figure S2: GC-MS chromatography of fresh tea leaves, green tea, oolong tea and black tea, Figure S3: GC-FID chromatography of fresh tea leaves, green tea, oolong tea and black tea, Figure S4: Volatile distributions among fresh tea leaves, green tea, oolong tea, and black tea, Table S1: Tea volatile FID response factor relative to ethyl caprate.

**Author Contributions:** Conceptualization, M.C.; methodology, M.C.; validation, H.Z., M.S. and W.Z.; formal analysis, M.C.; investigation, L.G.; resources, Z.L. and H.W.; data curation, L.G.; writing—original draft preparation, M.C.; writing—review and editing, W.Z., and H.W.; visualization, Y.Z.; supervision, Y.G.; project administration, M.C. and Y.G.; funding acquisition, H.W. All authors have read and agreed to the published version of the manuscript.

**Funding:** This research was supported by the Key-Area Research and Development Program of Guangdong Province (2020B020220004), the Science and Technology Innovation Fund of Fujian Agriculture and Forestry University (CXZX2017350), the Science and Technology Innovation Project of Chinese Academy of Agricultural Sciences (CAAS-ASTIP-2014-TRICAAS), and the Earmarked Fund for China Agricultural Research System (CARS-19).

**Institutional Review Board Statement:** Not applicable.

**Informed Consent Statement:** Not applicable.

**Data Availability Statement:** The data presented in this study are available in Supplementary Material here.

**Acknowledgments:** The authors want to thank Fuqing Wuli Ecological Agriculture Science and Technology Ltd. for providing the fresh tea leaves and graduate students Wei Wang and Zan Wang for helping with tea making.

**Conflicts of Interest:** The authors declare no conflict of interest. The funders had no role in the design of the study; in the collection, analyses, or interpretation of data; in the writing of the manuscript, or in the decision to publish the results.

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
