# Peer review of "Absolute Quantitative Volatile Measurement from Fresh Tea Leaves and the Derived Teas Revealed Contributions of Postharvest Synthesis of Endogenous Volatiles for the Aroma Quality of Made Teas"

_applsci, doi:10.3390/app11020613_

Round 1
Reviewer 1 Report
This article describes the investigation of aroma compounds in green tea, oolong tea, and black tea in order to determine how much the aroma compounds of fresh tea impact the quality of made teas. The manuscript is well-writing (especially as English may not be the authors first language), the purpose clear, and the work well organized. I am sure this work will be of interest to the tea industry. I have some minor corrections and a couple of suggestions:
Suggestion:
-in section 2.5 (lines 124 and 125) the statistical analysis is described. I would consider moving the first sentence to the table legend. Why is n 3 or 4? Maybe indicate the n value in the table. Also, the second sentence states that student t-tests were performed. I don’t see in the manuscript how results from these tests were used. Perhaps statistically different values in the table could be indicated in some way.
-in would be nice to add aroma descriptions (if known) in Table 1.
Corrections:
- Odor-activity value is abbreviated to OVA in several places (lines 64, 196, 207) and OAV in line 184. This should be consistent (presumably OAV is correct).
-line 84 “about 75%”. Not very scientific. Can you be more precise? E.g. “varies between x% and y%” with a reference or description of method used to determine?
-line 86. Replace “grounded” with “ground”.
-line 88. “appropriate amount of anhydrous sodium sulfate”. How was this determined?
-line 91. “concentrated to 500 µL”. How was this achieved?
-line 99. “zero air”. What does this mean? Please explain.
-line 103. “authentic standards”. Please provide details.
-line 129. Just a question. Should this be a semicolon? E.g. “total; 31, 32…”
-line 132. By convention, sentences normally do not begin with a number.
-line 166. Space between “these” and “volatile”
-line 193. Rewrite. E.g. “, which were NEVS produced in sufficient amount during tea processing, become aroma contributors.”
Author Response
Review 1 comments:
This article describes the investigation of aroma compounds in green tea, oolong tea, and black tea in order to determine how much the aroma compounds of fresh tea impact the quality of made teas. The manuscript is well-writing (especially as English may not be the authors first language), the purpose clear, and the work well organized. I am sure this work will be of interest to the tea industry. I have some minor corrections and a couple of suggestions:
Authors:Thank you very much for your encouraging words!
Suggestion:
-in section 2.5 (lines 124 and 125) the statistical analysis is described. I would consider moving the first sentence to the table legend. Why is n 3 or 4? Maybe indicate the n value in the table. Also, the second sentence states that student t-tests were performed. I don’t see in the manuscript how results from these tests were used. Perhaps statistically different values in the table could be indicated in some way.
Author: The original lines 124-125 were moved to the bottom of Table 1. The biological replicates were indicted in Table 1. The statistically significant difference was indicated by different upper case letters from same row and explained in the bottom of Table 1.
-in would be nice to add aroma descriptions (if known) in Table 1.
Authors: The aroma descriptions were added into Table 1 as one column.
Corrections:
- Odor-activity value is abbreviated to OVA in several places (lines 64, 196, 207) and OAV in line 184. This should be consistent (presumably OAV is correct).
Authors: The "OVA" were revised to "OAV" (line 69, 231 and 242).
-line 84 “about 75%”. Not very scientific. Can you be more precise? E.g. “varies between x% and y%” with a reference or description of method used to determine?
Authors: Revised as “between 65% to 75% depending on the tenderness of the tea leaves, germplasms and plucking seasons” (lines 94-95).
-line 86. Replace “grounded” with “ground”.
Authors: revised as suggested (line 96)
-line 88. “appropriate amount of anhydrous sodium sulfate”. How was this determined?
Authors: revised as “To absorb water, 8 g (for fresh tea leaves) or 0.5 g (for dry tea) anhydrous sodium sulfate were also added” (lines 998-100)
-line 91. “concentrated to 500 µL”. How was this achieved?
Authors: The volatile distillate was collected, aliquot into 8-mL glass tube, and concentrated in rotary evaporator to about 1.5 mL, then transferred into labeled 2-mL GC vial, further concentrated in rotary evaporator. The volume was closely monitored; the concentration step was stopped right way once the volume slightly below the 500 μL marker line. A small volume of diethyl ether was added to make the total volume to 500 μL. (lines 104-108).
-line 99. “zero air”. What does this mean? Please explain.
Authors: Zero air is a highly purified air in which almost all the hydrocarbon is removed (line 119-120).
-line 103. “authentic standards”. Please provide details.
Authors: we double-checked our chemical identification work flow, which was based on MS spectra match with NIST14 database and reported retention index. Thus, these two words were removed (line 125-126).
-line 129. Just a question. Should this be a semicolon? E.g. “total; 31, 32…”
Authors: revised as suggest (line 158).
-line 132. By convention, sentences normally do not begin with a number.
Authors: revised as “there were 19, 15, and 14…..” (lines 159-162)。
-line 166. Space between “these” and “volatile”
Authors: space added (line 196).
-line 193. Rewrite. E.g. “, which were NEVS produced in sufficient amount during tea processing, become aroma contributors.”
Authors: Revised as suggested (lines 225-226), thanks!

Reviewer 2 Report
The work is well constructed, and even if the topic is not new and much treated, the results add to and complete the already existing literature data.
Author Response
The work is well constructed, and even if the topic is not new and much treated, the results add to and complete the already existing literature data.
Authors: Thanks lot for your comments!
Reviewer 3 Report
General comment:
The present communication deals with the quantitation of volatile contents in fresh tea leaves, green tea, oolong tea and black tea. The manuscript is well written in general, although, in my opinion, important revision and additional experiments should be conducted before consideration for publication. It would be very interesting to briefly compare this new work with results obtained in other works from literature, e.g., Journal of Chromatography A, 1313 (2013) 245– 252. Although the authors have included this reference in the manuscript, they use it only in the introduction to explain the classification of tea categories. In my opinion, this paper in particular is not a good reference to understand the different classes of tea. Authors should change this. In addition, it is not clear whether they use either GC-MS, GC-FID or both methodologies together for quantification. This should be better explained and specified in the text (section 2.3). Finally, I consider that a figure comparing the chromatograms of fresh tea leaves, green tea, oolong tea and black tea should be added to the manuscript.
Specific comments:
Keywords:
- Remove GC-MS. In the present manuscript, this work is mainly based on GC-FID rather than GC-MS.
Introduction:
- Line 40: Reference #1 is not a valid reference to understand the different classes of tea. It should be changed for a better reference.
- Line 46: change “… but present from made teas.” to “… but present in made teas”.
- Line 63: change “…lower than that of the fresh tea leaves.” to “…lower than those of the fresh tea leaves.”
- Lines 64 (also 196 (twice), and 207): change “OVA” to “OAV”. Revise again the text, tables and figures for other possible typographical error.
Materials and methods
- Line 67: change “Materials and Methods” to “Materials and methods”
- Line 68: section 2.1 should be “Sample and chemicals”. Information about different chemicals, solvents, etc., regarding purity, suppliers and other important information should be included here.
- Line 69 – 82: it is not clear how the “heat deactivation” is performed. How is temperature controlled? Is it done in a simple oven? A special chamber? Also, is humidity controlled during the process? What is the pressure? Atmospheric?
- Line 86: what does it mean “grounded in the presence of liquid nitrogen”? I suppose they mean “ frozen with liquid nitrogen and powdered or grounded”. Please rewrite.
- Line 88: what is an “appropriate amount”? At least, an interval of the used amount should be specified in order to be able to replicate the work.
- Line 89: why 2 hours? Has this time been optimised? Explain.
- Line 87 - 92: conditions regarding solvent-assisted-flavor-evaporation should be better explained. For instance, what is the volume of the flasks used?. How do the authors concentrate the distillate afterwards? What pressure do they use?
- Line 98: the temperatures of injector and ion source seem to be interchanged. However, if it is so, the temperature of the injector does not seem to be correct if it is 50°C, since, if the oven temperature program starts at 50°C and raises to 180°C, it seems unlikely that the analytes can get out of the injector just with 50°C, without further heating. Perhaps it was 150°C?
- Lines 99-100: remove the hyphens from all figures.
- Line 103: what are the “authentic standards”? How were they measured? Explain here and add all the corresponding information in section “Sample and chemicals”.
- Line 113: change “The calculation of…” to “Calculation of …”
- Lines 118-122: the authors make what in my opinion is a very important assumption here, without further explanation or a literature base. This is an important part of the work and should be much better explained. Is there any way to determine how much of the volatiles from dry tea are extracted into the infusion? Why this assumption?
- Line 124: it is “A Student’s t-test”.
Results and discussion
- Line 130: the value given for the comparison between fresh tea leaves and black tea (13) does not correspond to what is shown in Figure S2. According to the figure, it should be 14. Revise and correct.
- Line 148: here, the “DW” abbreviation is used for the first time in the manuscript and it is not clarified what it means. I suppose it is “dry weight”. Explain and add the corresponding text.
- Lines 157 – 172: here, the authors make several explanations regarding volatility and evaporation. A table with the boiling points of the volatiles analysed should be included in the manuscript, perhaps, as an additional column in Table 1.
- Line 158: I think that the assertion about the loss of volatiles due to drying at 90-100°C is obvious. This was predictable.
- Line 166: there is a blank space missing between “these-volatile” at the end of the line.
- Lines 181-183: a short explanation of what is odor activity value, how it is calculated and most importantly, a proper literature reference should be included here.
- Lines 195-196: authors should perform a recovery and repeatability study of the proposed methodology. Also, the authors should give values for limits of detection and quantification. In general, validation of the method should be added and explained.
- Line 204: I am not very confident with the validity of the website used as a reference here. In addition, the format of the reference “in text” does not seem appropriate. Move to the “References” section.
- Line 207: explain the order of analytes used to write Table 2.
Conclusions
General comment: although the authors use GC-MS in keywords and give some information about it in section 2.3, they do not seem to use it for quantification at the end. Even in this section, they do not comment anything about it. A comment about the use of GC-MS should be included here or remove all GC-MS references from the paper and only explain GC-FID.
- Line 214: change “… tea from same batch…” to “… tea from the same batch…”
Figure captions
- Figure S1: Change “The standard tea making processes…” to “Standard tea making processes…”
References
The authors do not use the correct format of the journal “1. Author 1, A.B.; Author 2, C.D. Title of the article. Abbreviated Journal Name Year, Volume, page range.” This should be corrected.
Tables
- Table 1: include the meanings of “RT”, “RIn", “RIexp”, and all super indexes “a”, “b”, “c” at the foot of the table.
- Table 1: the “Total” value given at the end should be expressed with the proper significant figures.
- Table 2: What is the order of analytes used in this table?
- Table 2: Remove references from table foot and add the meaning of “nd” and “nf”.
- Table 2: the text size is not the same in some columns and rows.
- Table S1: What is the order of analytes used in this table?
Figures
- Figure S1: reduce the three boxes of “Solar withering” to just one. It is repetitive.
- Figure S2: consider the comment for line 130 and revise.
Author Response
Reviewer 3:
The present communication deals with the quantitation of volatile contents in fresh tea leaves, green tea, oolong tea and black tea. The manuscript is well written in general, although, in my opinion, important revision and additional experiments should be conducted before consideration for publication. It would be very interesting to briefly compare this new work with results obtained in other works from literature, e.g., Journal of Chromatography A, 1313 (2013) 245– 252. Although the authors have included this reference in the manuscript, they use it only in the introduction to explain the classification of tea categories. In my opinion, this paper in particular is not a good reference to understand the different classes of tea. Authors should change this. In addition, it is not clear whether they use either GC-MS, GC-FID or both methodologies together for quantification. This should be better explained and specified in the text (section 2.3). Finally, I consider that a figure comparing the chromatograms of fresh tea leaves, green tea, oolong tea and black tea should be added to the manuscript.
Authors: thank you very much for your thoughtful comments! We added comparisons between this work and the work of Zhang’s (2013) (line 248-261). The volatiles showing similar and diverse trends among green tea, oolong tea and black tea between these two studies were identified and briefly discussed.
New references were cited to replace the original reference 1.
We used GC-MS for volatile identification, and GC-FID for quantification. Explanations were listed in lines 123-126.
Two figures comparing the chromatograms of fresh tea leaves, green tea, oolong tea and black tea were added as Figure S2 and Figure S3.
Specific comments:
Keywords:
Remove GC-MS. In the present manuscript, this work is mainly based on GC-FID rather than GC-MS.
Authors: done as suggested (line 38).
Introduction:
Line 40: Reference #1 is not a valid reference to understand the different classes of tea. It should be changed for a better reference.
Authors: the original reference 1 was replaced by two new references (line 45).
Line 46: change “… but present from made teas.” to “… but present in made teas”.
Authors: corrected (line 51).
Line 63: change “…lower than that of the fresh tea leaves.” to “…lower than those of the fresh tea leaves.”
Authors: corrected (line 68).
Lines 64 (also 196 (twice), and 207): change “OVA” to “OAV”. Revise again the text, tables and figures for other possible typographical error.
Authors: Corrected (line 69, 231, and 242), we also double-checked all tables for other possible typographical error, thank you!
Materials and methods
Line 67: change “Materials and Methods” to “Materials and methods”
Authors: corrected (line 72).
Line 68: section 2.1 should be “Sample and chemicals”. Information about different chemicals, solvents, etc., regarding purity, suppliers and other important information should be included here.
Authors: Revised (line 73). Chemical information was added (lines 89-91).
Line 69 – 82: it is not clear how the “heat deactivation” is performed. How is temperature controlled? Is it done in a simple oven? A special chamber? Also, is humidity controlled during the process? What is the pressure? Atmospheric?
Authors: Heat deactivation was conducted in an electric-powered pan, the leaf temperature was monitored, and the deactivation was stopped once leaf temperature reached to 80-85°C (lines 82-84).
Line 86: what does it mean “grounded in the presence of liquid nitrogen”? I suppose they mean “frozen with liquid nitrogen and powdered or grounded”. Please rewrite.
Authors: revise as suggested (line 96).
Line 88: what is an “appropriate amount”? At least, an interval of the used amount should be specified in order to be able to replicate the work.
Authors: we add 8 g anhydrous sodium sulfate to 8g fresh tea leaves and 0.5 g anhydrous sodium sulfate to 2 g of dry tea (lines 99-100).
Line 89: why 2 hours? Has this time been optimised? Explain.
Authors: In an initial experiment we compared different extraction times (1 h, 2 h, and 12 h), we found that longer than 2h extraction was unnecessary. Other extraction optimization studies were reported previously (Chen et al., 2020, Flavour and Fragrance J. DOI: 10.1002/ffj.3617).
Line 87 - 92: conditions regarding solvent-assisted-flavor-evaporation should be better explained. For instance, what is the volume of the flasks used?. How do the authors concentrate the distillate afterwards? What pressure do they use?
Authors: the distillation flask and the receiver flask was 500 mL. The volatile distillate was collected, aliquot into 8-mL glass tube, and concentrated in rotary evaporator to about 1.5 mL, then transferred into labeled 2-mL GC vial, further concentrated in rotary evaporator. The volume was closely monitored; the concentration step was stopped right way once the volume slightly below the 500 μL marker line. A small volume of diethyl ether was added to make the total volume to 500 μL (lines 102-108).
Line 98: the temperatures of injector and ion source seem to be interchanged. However, if it is so, the temperature of the injector does not seem to be correct if it is 50°C, since, if the oven temperature program starts at 50°C and raises to 180°C, it seems unlikely that the analytes can get out of the injector just with 50°C, without further heating. Perhaps it was 150°C?
Authors: Sorry for this typo, the injector and ion source temperature we set in this study were 250°C and 230°C, respectively (line 116). Thanks lot for pointing out this error!
Lines 99-100: remove the hyphens from all figures.
Authors: done as suggested (lines 117-118).
Line 103: what are the “authentic standards”? How were they measured? Explain here and add all the corresponding information in section “Sample and chemicals”.
Authors: we double-checked our chemical identification workflow, which was based on spectral match with NIST14 database and reported retention index. These two words were removed (lines 125-126).
Line 113: change “The calculation of…” to “Calculation of …”
Authors: revised as suggested (line 136).
Lines 118-122: the authors make what in my opinion is a very important assumption here, without further explanation or a literature base. This is an important part of the work and should be much better explained. Is there any way to determine how much of the volatiles from dry tea are extracted into the infusion? Why this assumption?
Authors: To reliably extract the volatiles from the simulated standard tea infusion preparation is technically challenging, because volatiles are easily evaporated into air space and lost during tea infusion preparation; to add this complexity, it has been reported that during tea infusion preparation, some new volatiles can be formed from its glycosidically bound precursors under the action of hot water (Kinoshita et al., Food Chemistry 123 (2010) 601–606). To avoid these issues, we decided to calculate the aroma concentration in tea infusion under some reasonable assumptions (lines 140-146).
Line 124: it is “A Student’s t-test”.
Authors: corrected (line 153).
Results and discussion
Line 130: the value given for the comparison between fresh tea leaves and black tea (13) does not correspond to what is shown in Figure S2. According to the figure, it should be 14. Revise and correct.
Authors: Corrected (line 159).
Line 148: here, the “DW” abbreviation is used for the first time in the manuscript and it is not clarified what it means. I suppose it is “dry weight”. Explain and add the corresponding text.
Authors: added (line 178).
Lines 157 – 172: here, the authors make several explanations regarding volatility and evaporation. A table with the boiling points of the volatiles analysed should be included in the manuscript, perhaps, as an additional column in Table 1.
Authors: The boiling points were added to revised table 1 as an additional column, thanks!
Line 158: I think that the assertion about the loss of volatiles due to drying at 90-100°C is obvious. This was predictable.
Authors: agreed, thanks!
Line 166: there is a blank space missing between “these-volatile” at the end of the line.
Authors: blank space added (line 196-197).
Lines 181-183: a short explanation of what is odor activity value, how it is calculated and most importantly, a proper literature reference should be included here.
Authors: Explanation for OAV was added, and how it is calculated, and literature was added (lines 212-215).
Lines 195-196: authors should perform a recovery and repeatability study of the proposed methodology. Also, the authors should give values for limits of detection and quantification. In general, validation of the method should be added and explained.
Authors: Previously we performed a systematic study to optimize the methodology, and the results were reported (Chen et al., 2020, Flavor and Fragrance J. DOI: 10.1002/ffj.3617). Thus, in this study we did not duplicate these experiments.
Line 204: I am not very confident with the validity of the website used as a reference here. In addition, the format of the reference “in text” does not seem appropriate. Move to the “References” section.
Authors: New literatures were cited to replace the website (lines 239, 241), thanks!
Line 207: explain the order of analytes used to write Table 2.
Authors: The analytes were arranged by their retention time from low to high, and explanation was added in lines 216.
Conclusions
General comment: although the authors use GC-MS in keywords and give some information about it in section 2.3, they do not seem to use it for quantification at the end. Even in this section, they do not comment anything about it. A comment about the use of GC-MS should be included here or remove all GC-MS references from the paper and only explain GC-FID.
Authors: A comment about the use of GC-MS was added into section 2.3 (lines 121-123), thank you!
Line 214: change “… tea from same batch…” to “… tea from the same batch…”
Authors: revised as suggested (line 263).
Figure captions
Figure S1: Change “The standard tea making processes…” to “Standard tea making processes…”
Authors: revised as suggested.
References
The authors do not use the correct format of the journal “1. Author 1, A.B.; Author 2, C.D. Title of the article. Abbreviated Journal Name Year, Volume, page range.” This should be corrected.
Authors: the references were reformatted according to journal style, thanks!
Tables
Table 1: include the meanings of “RT”, “RIn", “RIexp”, and all super indexes “a”, “b”, “c” at the foot of the table.
Authors: Added as suggested, thanks!
Table 1: the “Total” value given at the end should be expressed with the proper significant figures.
Authors: The total volatile contents from fresh leaves and derived teas were expressed as Figure 1.
Table 2: What is the order of analytes used in this table?
Authors: the analytes were ordered based on their retention time from low to high.
Table 2: Remove references from table foot and add the meaning of “nd” and “nf”.
Authors: done as suggested.
Table 2: the text size is not the same in some columns and rows.
Authors: adjusted to make all font size even.
Table S1: What is the order of analytes used in this table?
Authors: the analytes were ordered by retention time from low to high.
Figures
Figure S1: reduce the three boxes of “Solar withering” to just one. It is repetitive.
Authors: The Figure S1 was modified as suggested to remove redundancy.
Figure S2: consider the comment for line 130 and revise.
Authors: double-checked, thanks!
Round 2
Reviewer 3 Report
Thank you for considering my comments and making the corresponding corrections.
Only a minor correction should be done to table 1, adding the meaning of "nd" and "no" at foot note. Also, reference(s) for boiling points in table 1 should be added here too.
Author Response
Dear editor,
We made revision to table 1 and the text file according to your comments. We tried to satisfy reviewer's comment to add references for the boiling point. However, we did not find one that systematically document these data. Currently, the boiling point that we provided in the table 1 was based on internet search, arguably which is not an appropriate resource. To add this complexity, most reported boiling points are based on pure chemical form, it's difficult to know their true boiling point in a mixture or when they interact with other matrix such as tea. Please feel free to let me know what I can do further, thank you!
Best regards,
Mingjie Chen, PhD, professor
College of Life Sciences
Xinyang Normal University
237 Nanhu Road,
Xinyang, Henan, 464000, China
Email: mjchen@xynu.edu.cn
Dear editor,
We made revision to table 1 and the text file according to your comments. We tried to satisfy reviewer's comment to add references for the boiling point. However, we did not find one that systematically document these data. Currently, the boiling point that we provided in the table 1 was based on internet search, arguably which is not an appropriate resource. To add this complexity, most reported boiling points are based on pure chemical form, it's difficult to know their true boiling point in a mixture or when they interact with other matrix such as tea. Please feel free to let me know what I can do further, thank you!
Best regards,
Mingjie Chen, PhD, professor
College of Life Sciences
Xinyang Normal University
237 Nanhu Road,
Xinyang, Henan, 464000, China
Email: mjchen@xynu.edu.cn
Dear editor,
We made revision to table 1 and the text file according to your comments. We tried to satisfy reviewer's comment to add references for the boiling point. However, we did not find one that systematically document these data. Currently, the boiling point that we provided in the table 1 was based on internet search, arguably which is not an appropriate resource. To add this complexity, most reported boiling points are based on pure chemical form, it's difficult to know their true boiling point in a mixture or when they interact with other matrix such as tea. Please feel free to let me know what I can do further, thank you!
Best regards,
Mingjie Chen, PhD, professor
College of Life Sciences
Xinyang Normal University
237 Nanhu Road,
Xinyang, Henan, 464000, China
Email: mjchen@xynu.edu.cn
Dear editor,
We made revision to table 1 and the text file according to your comments. We tried to satisfy reviewer's comment to add references for the boiling point. However, we did not find one that systematically document these data. Currently, the boiling point that we provided in the table 1 was based on internet search, arguably which is not an appropriate resource. To add this complexity, most reported boiling points are based on pure chemical form, it's difficult to know their true boiling point in a mixture or when they interact with other matrix such as tea. Please feel free to let me know what I can do further, thank you!
Best regards,
Mingjie Chen, PhD, professor
College of Life Sciences
Xinyang Normal University
237 Nanhu Road,
Xinyang, Henan, 464000, China
Email: mjchen@xynu.edu.cn
Dear editor,
We made revision to table 1 and the text file according to your comments. We tried to satisfy reviewer's comment to add references for the boiling point. However, we did not find one that systematically document these data. Currently, the boiling point that we provided in the table 1 was based on internet search, arguably which is not an appropriate resource. To add this complexity, most reported boiling points are based on pure chemical form, it's difficult to know their true boiling point in a mixture or when they interact with other matrix such as tea. Please feel free to let me know what I can do further, thank you!
Best regards,
Mingjie Chen, PhD, professor
College of Life Sciences
Xinyang Normal University
237 Nanhu Road,
Xinyang, Henan, 464000, China
Email: mjchen@xynu.edu.cn
Dear editor,
We made revision to table 1 and the text file according to your comments. We tried to satisfy reviewer's comment to add references for the boiling point. However, we did not find one that systematically document these data. Currently, the boiling point that we provided in the table 1 was based on internet search, arguably which is not an appropriate resource. To add this complexity, most reported boiling points are based on pure chemical form, it's difficult to know their true boiling point in a mixture or when they interact with other matrix such as tea. Please feel free to let me know what I can do further, thank you!
Best regards,
Mingjie Chen, PhD, professor
College of Life Sciences
Xinyang Normal University
237 Nanhu Road,
Xinyang, Henan, 464000, China
Email: mjchen@xynu.edu.cn
Dear editor,
We made revision to table 1 and the text file according to your comments. We tried to satisfy reviewer's comment to add references for the boiling point. However, we did not find one that systematically document these data. Currently, the boiling point that we provided in the table 1 was based on internet search, arguably which is not an appropriate resource. To add this complexity, most reported boiling points are based on pure chemical form, it's difficult to know their true boiling point in a mixture or when they interact with other matrix such as tea. Please feel free to let me know what I can do further, thank you!
Best regards,
Mingjie Chen, PhD, professor
College of Life Sciences
Xinyang Normal University
237 Nanhu Road,
Xinyang, Henan, 464000, China
Email: mjchen@xynu.edu.cn
Dear editor,
We made revision to table 1 and the text file according to your comments. We tried to satisfy reviewer's comment to add references for the boiling point. However, we did not find one that systematically document these data. Currently, the boiling point that we provided in the table 1 was based on internet search, arguably which is not an appropriate resource. To add this complexity, most reported boiling points are based on pure chemical form, it's difficult to know their true boiling point in a mixture or when they interact with other matrix such as tea. Please feel free to let me know what I can do further, thank you!
Best regards,
Mingjie Chen, PhD, professor
College of Life Sciences
Xinyang Normal University
237 Nanhu Road,
Xinyang, Henan, 464000, China
Email: mjchen@xynu.edu.cn